# Omega-3 Lipid Mediators: Modulation of the M1/M2 Macrophage Phenotype and Its Protective Role in Chronic Liver Diseases

**DOI:** 10.3390/ijms242115528

**Published:** 2023-10-24

**Authors:** Luis Alberto Videla, Rodrigo Valenzuela, Andrea Del Campo, Jessica Zúñiga-Hernández

**Affiliations:** 1Molecular and Clinical Pharmacology Program, Institute of Biomedical Science, Faculty of Medicine, University of Chile, Santiago 8380000, Chile; lvidela1944@gmail.com; 2Nutrition Department, Faculty of Medicine, University of Chile, Santiago 8380000, Chile; rvalenzuelab@uchile.cl; 3Laboratorio de Fisiología y Bioenergética Celular, Escuela de Química y Farmacia, Facultad de Química y de Farmacia, Pontificia Universidad Católica de Chile, Santiago 8331150, Chile; andreadelcamposfeir@gmail.com; 4Biomedical Sciences Department, Faculty of Health Sciences, University of Talca, Talca 3460000, Chile

**Keywords:** chronic liver disease, specialized pro-resolving mediators, inflammation, macrophages M1/M2 polarization, immunometabolism

## Abstract

The complex interplay between dietary factors, inflammation, and macrophage polarization is pivotal in the pathogenesis and progression of chronic liver diseases (CLDs). Omega-3 fatty acids (FAs) have brought in attention due to their potential to modulate inflammation and exert protective effects in various pathological conditions. Omega-3 fatty acids eicosapentaenoic acid (EPA) and docosahexaenoic acid (DHA) have shown promise in mitigating inflammation and enhancing the resolution of inflammatory responses. They influence the M1/M2 macrophage phenotype balance, promoting a shift towards the M2 anti-inflammatory phenotype. Specialized pro-resolving mediators (SPMs), such as resolvins (Rvs), protectins (PDs), and maresins (MaRs), have emerged as potent regulators of inflammation and macrophage polarization. They show anti-inflammatory and pro-resolving properties, by modulating the expression of cytokines, facilitate the phagocytosis of apoptotic cells, and promote tissue repair. MaR1, in particular, has demonstrated significant hepatoprotective effects by promoting M2 macrophage polarization, reducing oxidative stress, and inhibiting key inflammatory pathways such as NF-κB. In the context of CLDs, such as nonalcoholic fatty liver disease (NAFLD) and cirrhosis, omega-3s and their SPMs have shown promise in attenuating liver injury, promoting tissue regeneration, and modulating macrophage phenotypes. The aim of this article was to analyze the emerging role of omega-3 FAs and their SPMs in the context of macrophage polarization, with special interest in the mechanisms underlying their effects and their interactions with other cell types within the liver microenvironment, focused on CLDs and the development of novel therapeutic strategies.

## 1. Introduction

Historically, high fat consumption in humans has been associated with metabolic problems, metabolic disease deterioration, overweight, and cardiovascular disease. However, since 1960, there has been increasing evidence of the health benefits and protective effects of omega-3 fatty acids (FAs) [1,2]. This has led to physiological resolution of inflammation and could have a therapeutic role in a wide range of pathologies, both acute and chronic, among the latter being chronic liver diseases (CLDs) [2,3]. According to the global burden of liver disease: 2023 update, CLD was the 11th cause of mortality and the 15th cause of disability-associated life-years worldwide [4,5]. In the context of CLD associated with metabolic disorders, the nonalcoholic fatty liver disease (NAFLD) has increased in the last years, the overall prevalence of NAFLD worldwide is 32.4%, and may appear in absence if obesity or metabolic syndrome clinical criteria’s, and may coexist with insulin resistance or cardiovascular risk [4]. According to the Center for Disease Control and Prevention Analysis (CDC), the NAFLD cases in man are expected to increase by 21% in the US for the 2015–2030 period, a pattern that will be followed by other occidental countries [6]. These records justify the search for new non-invasive treatments with a better cost/effectiveness ratio. In this context are the recently discovered specialized proresolving mediators (SPMs), among which are lipoxins from omega-6 polyunsaturated FAs (PUFAs) and resolvins, protectins, and maresins from omega-3 PUFAs, synthesized by polymorphonuclear leukocytes (PMNs) [7]. SPMs are able to reduce inflammatory infiltration, and promote phagocytosis of cellular debris and tissue regeneration, in addition of having analgesic and anti-inflammatory effects [7]. Due to their role in anti-inflammation, SPMs could be associated with a positive action over the resident macrophages of the liver, the Kupffer cells (KCs), and circulating monocyte-derived macrophages (MoMFs) in murine models [8]. KCs are crucial in maintaining an inflammatory microenvironment, characteristic of the development of cirrhosis. According to these antecedents, the aim of this review article was to examine the influence of n-3 PUFAs and their SPM derivates in the transformation of KCs expressing the proinflammatory phenotype M1 towards the pro-resolving or immune-modulating phenotype M2, which regulates late repair processes, resolution of inflammation and induction of immune tolerance, necessary steps at the beginning of tissue regeneration [9,10].

## 2. Chronic Liver Disease

CLD is any long-standing condition that prevents the normal functioning of the liver [11]. Cirrhosis can be a consequence of different causes, such as obesity, NAFLD, high alcohol consumption, and hepatitis B or C virus infection, among others. Cirrhosis is an irreversible and diffuse process of liver damage developed after a long period of inflammation that results in replacement of the healthy liver parenchyma by fibrotic tissue and regenerative nodules, with collapse and distortion of liver structures [12,13].

## 3. Liver Inflammation

Hepatocytes are cells that are highly resistant to variations in their environment, but they can undergo a series of reversible changes, among which are the accumulation of fat or steatosis, and that of bilirubin or cholestasis. Furthermore, when the damage exceeds the homeostatic mechanisms of the hepatocyte, cell death ensues, which may occur due to necrosis, apoptosis [11], or ferroptosis [12]. When a considerable amount of tissue becomes necrotic, the space where there should be the hepatocytes is replaced by cellular debris. The inflammatory cytokines derived from this process attract PMNs and activate hepatic stellated cells (HSCs), making them acquire a myofibroblast phenotype that increases their proliferation and fibrogenesis [13]. Under normal conditions, the extracellular matrix (ECM) of Disse’s space is composed of type I, III, and IV collagens, fibronectin, laminin, heparan sulfate, and dermatan sulfate. As the noxious stimulus is prolonged, the production of HSCs will alter their proportions, increasing the total amount of ECM by three to five times. This alteration favors tissue dysfunction, since under normal conditions, the ECM components interact with neighboring cells, regulating their shape, motility, survival, differentiation, and gene expression [13,14]. The excessive production of ECM evolves into fibrous septa. In CLD these septa surround clusters of regenerating hepatocytes forming diffuse scarring or cirrhosis. Thus, fibrosis can be considered as an indicator of the extent of chronic damage. It is important to note that under experimental conditions of hepatic fibrosis, the liver can return to a normal state upon removal of the harmful stimuli, a process known as fibrosis resolution (reversion of HSCs activation) [14,15,16].

## 4. Liver Macrophages

Macrophages are immune cells described by Metchnikoff, and received their name thanks to Aschoff’s studies, as part of the reticule-endothelial system [17]. It is now recognized that murine macrophages originate from monocyte recruitment dependent and independent mechanisms, originated and established during embryonic development) [18]. The origin of macrophages dates back to the early stages of pregnancy, where primitive hematopoiesis occurs in the extra-embryonic yolk sac, which has as its exclusive product red blood cells and macrophages, without a monocytic progenitor. In parallel, hematopoietic stem cells arise from the mesoderm, which, during embryogenesis, migrate to the fetal liver where temporary hematopoiesis occurs [8,19].

Liver-resident macrophages derive from colonies of the erythromyeloid yolk sac, which migrate to the liver in early stages of murine gestation, subsequently being capable of self-renewal without the need to depend on bone marrow progenitors [20], but the latter being able to differentiate into fully functional KCs, if necessary, as occurs in inflammatory processes. KCs are the largest reserves of resident macrophages, with 20% of non-parenchymal cells in the liver being located in the lumen of hepatic sinusoids, a location that favors their phagocytic function to support liver homeostasis [21,22]. Faced with chronic inflammatory stimuli, KCs change their tolerogenic phenotype towards an activated phenotype where, focusing their functions on the development of an inflammatory state that, added to the decrease in homeostatic functions, contribute to the development of hepatic pathological states [22]. Both KCs and circulating murine monocytes are characterized by express the cell surface glycoprotein F4/80^+^, recognized as a distinctive marker of murine macrophages [23]. In addition, the resident KCs mainly express the cluster of differentiation CD68^+^, but CD11b^+^ poorly, and have a strong phagocytic and bactericidal activity, in contrast to the macrophages derived from bone marrow expression high in CD11b^+^, which mainly produce cytokines (tumor necrosis factor-α (TNF-α), interleukin-12 (IL-12)), and that during inflammation accumulate and activate in the affected areas [23,24].

## 5. Macrophage Phenotypes M1 and M2

Under various stimuli, the macrophages are activated through two common described polarization ways, depending on the specific gene transcription cascade, namely, the classical pathway exhibiting the pro-inflammatory phenotype (M1 or relative to CD4^+^ T helper (Th1)), and the alternative immune-modulatory pathway (M2 or relative to CD4^+^ Th2) also denominated anti-inflammatory phenotype [25,26]. M2 macrophages are divided into M2a, M2b, M2c, and M2d subclasses, differing by their surface markers, cytokines secreted, and immune functions. The production of M1-derived pro-inflammatory cytokines initially regulates the processes of early repair and response to pathogens, while M2-derived cytokines such as IL-4, IL-10, and IL-13, among others, stimulate the late repair process, immunity resolution, and tolerance [27]. It is important to remark that this model of macrophage polarization is not exempted from controversy. The functions that traditionally are attributed to certain pathways or molecular patterns are limited, and there is a lack of tightly defined criteria to score phenotypes due to the complexity of the regulation. [10,28].

Among the most common stimulus for M1 phenotype is the interferon-*γ* (INF-*γ*) induced by Th-1, natural killer cells (NKs), and macrophages. The activation of INF-*γ* receptor (IFNGR 1 and 2) by INF-*γ* controls specific gene expression programs involving cytokine receptors (including IL-15 receptor α [RA], IL-2RA, and IL-6R); cell activation markers (CD36, CD38, CD69, and CD97); and a number of cell adhesion molecules (intercellular adhesion molecule 1 [ICAM1], integrin alpha L [ITGAL], mucin 1 [MUC1], and ST6 beta-galactosamide, among others) related to Th-1 response [29,30]. Other activators of M1 are the pathogen associated molecular patterns (PAMPs) such as lipopolysaccharide (LPS) and granulocyte-macrophage colony-stimulating factor (GM-CSF), all of them favoring antigen presentation, phagocytosis, chemotaxis, adhesion, and pro-inflammatory cytokine production [26,29]. Otherwise, one of the inducers of M2 (particularly M2a) is IL-4. The human and murine IL-4 favors the Krüppel-like factor 4 (KLF4) and the peroxisome pathway [31]. The interaction with the IL-4 receptor-1 (IL-4Rα) induces Bcl-2-like protein-1 (BCL2L1) and B-cell lymphoma-extra-large (BCLXL), leading to diminution of apoptosis, fusion of macrophages, promotion of transforming growth factor-β1 (TGF-β1), and production of transglutimase-2 (TGM2) as a co-receptor of peroxisome proliferator-activated receptor-*γ* (PPAR*γ*). PPAR*γ* appears in the late stages of polarization, regulating progressive and/or reinforcing macrophage polarization [29,32]. In the case of M2c, the glucocorticoid receptor (GCR) induces the formation of IL-1R2, IL-10, and CD163 and accelerates the autophosphorylation of IL-10R inhibiting the production of pro-inflammatory cytokines and the stimulation of chemoattractant factors such as CXC motif chemokine ligand 13 (CXCL-13) and CXCL4 [29]. Due to the M1/M2 equilibrium playing a role in inflammation, proliferation, and remodeling, the proportion of the phenotypes has been studied in various models of chronic diseases [33]. Of interest is the relation between inflammatory versus regeneration phenomena, and CLD has been associated with a misbalance in M1/M2 response [33].

## 6. Omega-3 Fatty Acids and Their Role on M1/M2 Macrophage

The study of omega-3s dates back to the early 1960s. Danish researchers Bang and Dyerberg observed that Alaska Natives with a diet high in omega-3s, derived from eating seal meat and fish, had a lower incidence of heart disease [1,34]. This study began a series of investigations in which the importance and benefits of this type of fat in the diet were studied. Omega-3s are found in three main forms, namely, *α*-linolenic acid (ALA, 18C:3 n-3), eicosapentaenoic acid (EPA, 20C:5 n-3), and docosahexaenoic acid (DHA, 22C:6 n-3). In food, EPA and DHA are found in fish such as salmon and tuna, or algae, while vegetable oils, chia, and nuts are the main foods with a high content of ALA [35]. ALA is a precursor of EPA and DHA, and the synthesis of both fatty acids occurs through desaturation and elongation reactions mainly in the liver, and in brain, testicle, and kidney, [36]. Also, EPA and DHA synthesis is directly related with nutritional status and oxidative stress [37]. Several studies show the importance of maintaining omega-6/omega-3 consumption in a ratio between 1/1 and 4/1. It should be considered that the average Western diet, heavily based on the consumption of meat and animal fats, has an approximate ratio of 16/1 in what the consumption of omega-6 and omega-3 refers to [38]. It has been observed that this results in increased metabolism of omega-6 polyunsaturated fatty acids (PUFAs), generating a greater number of metabolic mediators that, as mentioned above, have proinflammatory and neoplastic properties [39].

EPA and DHA (at 100 mM) have been shown to reduce oxidative damage to endothelial cell DNA, reducing the concentration of H_2_O_2_ and other reactive oxygen species (ROS) at the intracellular level in a model of human aortic endothelial cells (HAECs). These findings suggest that omega-3s have protective effects at the genetic level through mechanisms that reduce damage to genetic material without promoting its repair [40]. Based on the above, it was reported that the consumption of omega-3 is related to an improvement in endothelial function associated with an increase in nitric oxide (NO) in human endothelial cells [41,42]. In addition, it has been observed that the administration of omega-3 has protective functions in the development of some relevant pathologies (see Figure 1). In different animals, clinical trials, and cellular models, the treatment with DHA and/or EPA promotes beneficial effects through the activation of immune cells such as PMNs or monocytes, decreasing inflammation [3,43,44,45,46,47,48,49]. For example, the addition of 2.4 g/d to type two diabetes (T2DM) patients for 8 weeks may modulate the activity of PPAR*γ* nuclear receptors, protecting by this way the cardiovascular system against atherosclerotic lesion formation and exerting an anti-inflammatory role [46]. Following the same line, the administration of EPA+DHA at 2 g/d (for six months) can lower c-reactive protein (as marker of cardiovascular disease) in patients with end-stage renal disease, and 4 g/d (for one month in healthy volunteers) can ameliorate acute and chronic vascular inflammation, with a decrease in C–C motif chemokine ligand 2 (CCL2), a chemokine that enhances macrophage responses to pro-inflammatory stimuli [47,48]. In all the cases previously explained, the central positive actions of omega-3 are related to their ability to modify the inflammatory status.

Moreover, the direct effect of these fatty acids can be seen on the different tissues, such as cardiac tissue (chosen due to its high relevance) where EPA has been reported to reduce heartbeat through a decrease in membrane potential [50]. On the other hand, in cardiac fibroblast, the combination of EPA and DHA (added to c57bl/6 mice for 8 weeks as 1% of total dietary energy) inhibit TGF-*β* pathways decreasing fibrosis [51], and in humans, the addition of 4 g/d of omega-3 ethyl esters (Omega-Remodell clinical trial) showed a significant reduction in left ventricular end-systolic volume index, and high dose of omega-3 is associated with significant reduction in inflammation and myocardial fibrosis in patients during convalescent phase of acute infarct healing [43,52]. Also, it was found, in clinical trial analysis, that EPA plus DHA can decrease lipogenesis and liver steatosis [53,54]. Additionally, EPA and DHA being both preventive and therapeutic protective factors related to immunity and particularly macrophages anti-inflammatory balances, as shown in Table 1 [55,56,57,58,59,60,61,62,63,64,65,66,67]. The Table 1 presents a comprehensive summary of the effects of EPA/DHA on different mechanisms and models (clinical trials and animal studies). In clinical trials, EPA/DHA supplementation demonstrated promising results, such as reducing T helper 2/T helper 1 chemokines in newborns from mothers with pregnancy-related depression, improving various metabolic parameters in T2DM, and modulating inflammatory pathways in obese individuals. Moreover, it enhanced specific macrophage markers and promoted anti-inflammatory cytokine production in children with low DHA intake, and it contributed to improved atherosclerotic plaque morphology in patients awaiting carotid endarterectomy. In animal models, EPA/DHA exhibited a range of effects, including attenuating atherosclerotic plaque development and suppressing atherogenesis, reducing aneurysm formation and macrophage infiltration, modulating skin inflammation in a psoriasis model, and influencing various signaling pathways. These findings underscore the potential health benefits of EPA/DHA in diverse contexts, from cardiovascular health to inflammation modulation.

The importance of reviewing data from both human and animal trials in Table 1 allows us to make a critical comparison between the effects of EPA/DHA, where human clinical trials provide direct insights into the effects of EPA/DHA in human populations. The clinical trials focus on health conditions directly relevant to humans, such as pregnancy-related depression, T2DM and atherosclerosis in patients awaiting carotid endarterectomy. This relevance ensures that the observed effects have immediate implications for human health with high ethical standards to protect the rights and well-being of human participants. The wide range of health parameters studied, including chemokine levels, triglycerides, waist measurements, and atherosclerotic plaque morphology, underscores the potential multi-faceted benefits of EPA/DHA in human health. The phenomena observed in the clinical trial enable readily interpretations for medical practice and treatment strategies. For example, the reduction in triglyceride levels in T2DM patients suggests a potential therapeutic opportunity. Also, the results allow to understand that EPA/DHA can decrease inflammatory pathways mostly related to macrophages activity and infiltration, and T-cell cytokine production. Animal models, on the other hand, offer controlled environments for mechanistic studies and initial insights, but require cautious interpretation and translation to human health due to species differences in physiology and metabolism; however, preclinical studies allow elucidation of the underlying mechanisms of action, providing a deeper understanding of how EPA/DHA exert their effects at the cellular and molecular levels. For example, the Apolipoprotein E-deficient (ApoE^-/-^) mice model helps elucidate the role of EPA/DHA in atherosclerosis development and the induction of Nrf2 to promote the cytokines switch from pro- to anti-inflammatory response, offering valuable mechanistic insights into how EPA/DHA affects pathways, cellular processes, and disease development. For instance, the K14-Rac1V12 mouse model sheds light on EPA/DHA’s effects on psoriasis, due to an increase in the SPM in the skin. These findings serve as a starting point for further research in humans. Combining both types of research can yield a more comprehensive understanding of EPA/DHA’s effects, from basic mechanisms to clinical applications.

The effects of EPA and DHA are of great interest in the scientific community, both in the field of chronic diseases and in the polarization of the M1 and M2 macrophage phenotypes. In this sense, there are multiple studies on EPA and DHA that show their participation in the reduction in arachidonic acid (ARA)-derived lipid mediators and the modulation of inflammation. According to Allam-Ndoul et al. [68], a combination of 75 μM each of EPA and DHA in a 1:1 ratio modulates inflammation (dose-dependently) inhibiting polarization towards M1, in a human monocytic THP-1 cell line. EPA seems to have larger effects where inflammation is already established that it has resolving effects, while DHA and the EPA/DHA mixture presented higher effectiveness when administered at the same time as the proinflammatory inducer (LPS) [68]. Also, they found that DHA is more potent than EPA. In the same line, Kawano et al. [69] studied the effect of DHA on macrophage polarization, in cells of human monocytic lineage (U937 and THP-1). They showed a polarization towards M2 (CD23, CD206), with secretion of anti-inflammatory cytokines such as TGF-β and IL-10. DHA acts on a p38 MAPK-dependent pathway, a route that participates in the polarization and production of cytokines, and increases the transcription factor KLF4, an inhibitor of NF-κB activity [69]. In the context of metabolic disease, there are a few studies related to the role of omega-3 in macrophage polarization. Song et al. [70] found that macrophage infiltration into the liver and adipose tissue is diminished in a model of fat-1 high-fat diet (HFD)-induced obesity mice, where fat-1 is a gene modification able to convert omega-6 to omega-3 PUFAs in vivo. These investigators found that omega-3 suppresses proinflammatory M1, enhancing M2 polarization in adipose tissue macrophages, and causes anti-inflammatory and insulin-sensitizing effects {70]. More recently, Ontoria-Oviedo et al. reported that the administration of a commercial nutritional preparation of omega-3 (LIPINOVA^®^) could promote wound closure in a db/db model in the context of diabetes mellitus type-2 (DM2)-related ulcers [66]. The resolution of the wound was directly related to a decrease in the ratio M1/M2 in the area of the noxa, measured by the ratio F4/80^+^CD274^+^ (M1) /F4/80^+^/CD206^+^ (M2) [58]. Going further, Carpino et al. evaluated the macrophage polarization in 32 children with biopsy-proven NAFLD, 20 of whom received 250 mg/day DHA for 18 months [71]. DHA-treatment determined a significant reduction in liver steatosis, hepatocyte ballooning, and the number of portal CD68^+^ and total S100A9^+^ macrophages, concomitantly with the enhancement of the anti-inflammatories CD206^+^ and CD163^+^/Arginase1^+^ lobular macrophages. Also, DHA treatment caused an increased number of apoptotic macrophages [72]. Despite the slight information that exists at the hepatic level, there is clear evidence of factors that would be key in the modulation of the polarization of macrophages by omega-3s, and given their importance, the derivatives of these fatty acids will be described below.

## 7. Omega-3 Lipid Mediators

Studies in the last 15 years show the existence of specialized pro-resolving mediators (SPMs) derived from omega-3s with resolving and protecting properties in inflammatory processes, known as resolvins (Rvs), protectins (PDs/NPDs), and maresins (MaRs) [10,72,73]. These molecules were isolated for the first time from inflammatory exudates of murine models and are the product of a series of enzymatic reactions, where EPA and DHA are metabolized by the same enzymes, cyclooxygenase-2 (COX-2) and 15-lipoxygenase (15-LOX), that also participate in ARA metabolism. It should be noted that these enzymes have a higher affinity for ARA than omega-3s, which is why high concentrations of intracellular EPA and DHA are required for clinical benefits [74,75]. Among the Rvs, it is possible to find the E-series including RvE1 and RvE2. Their synthesis is initiated in the presence of aspirin (acetylsalicylic acid) which acetylates COX-2 that produces prostaglandins (PG) [76,77]. SPMs can also be generated independently of aspirin via cytochrome P450 mono-oxygenases that convert n-3 PUFAs into epoxy and hydroxy fatty acids [78,79]. The D-series of Rvs include RvD1/RvD2/RvD3/RvD4/RvD5/RvD6, and they can have or not forms derived from aspirin [80,81,82,83,84].

In relation to PD1 or Neuroprotectin (NPD1), their names change depending on where their biosynthesis occurs; the “neuro” suffix for NPD1 is added when it occurs in neural cells/neural ectoderm, while PD1 is mainly synthesized in immune cells [72,73,85]. In particular, PD1 (Table 2) has been related to a decrease in leukocyte infiltration in murine models of the immune, cardiovascular, and renal systems [79].

Maresin-1 and 2 were the last discovered SPMs. They are generated by the 12-LOX catalyzed epoxidation and enzymatic hydrolysis in macrophages and platelets [86,87,88,89]. In recent years, MaR1 has been studied in different tissues and organs, describing its resolution actions in inflammatory pathologies in rodent and human brain [90,91], heart [92,93], kidney [94,95], and liver [96,97,98] (Table 2). The effects of MaR1 would even exceed, in some cases, those produced by other pro-resolving molecules whose effects have already been proven, such as RvD1. This opens up the discussion of its efficacy in more complex and exhaustive models and its possible role, in the long term, as treatment of inflammatory diseases [89].

## 8. The SPMs in Relationship to the M1/M2 Macrophage Phenotype in Chronic Liver Diseases

The evidence indicates that EPA and DHA participate directly or not in the polarization of the macrophage phenotype [66,68,70]. Research in recent years indicates that lipid mediators with proinflammatory (eicosanoids) or anti-inflammatory (SPMs) characteristics are synthesized in compliance with the M1 and M2 macrophage phenotypes respectively [10,87,88,99] (see Figure 2). M2 macrophages produce MaR1 and lower levels of LTB4 and PG than M1 [88,99]. In the study of Sehan and Dalli [99], they observed that macrophages in efferocytosis feedback promoting the production of RvD3 and RvE1M. Studies carried out in 2011 by Titos et al. demonstrated that RvD1, in addition of being produced by macrophages of obese mice after high-fat diet feeding, increases the attenuated expression of IFNγ/LPS-induced Th1 cytokines (TNF-α and IL-6), while upregulating arginase 1 expression in a concentration-dependent manner [100,101]. These data are in agreement with that reported by Han et al., where MaR1 acts as an autoregulatory circuit through retinoid orphan receptor-α (RORα)/2-Lox activation [102]. In concordance with these findings, MaR1 inhibited an CD38^+^CD80^+^CD86^+^iNOS^+^ M1 macrophage differentiation promoting a CD36^+^CD163^+^CD206^+^Arg-1^+^ M2 in LPS-induced cardiac injury [103]. In the context of inflammatory pain, NPD1 as protectin conjugates in tissue regeneration 1 (PCTR1) increases phagocytic activity [104,105].

**Table 2 ijms-24-15528-t002:** Protective functions of n-3 PUFA derived specific proresolving mediators (SPMs; resolvins, protectins, and maresins) related to inflammatory and macrophage modulation.

**1. General Functions**		
**SPM**	**Doses**	**Effects**
Resolvin E1 (RvE1)	100 ng per mouse	Breaks off inflammatory infiltration by over 50% in a murine model of acute inflammation, promoting PMN macrophage ingestion, reducing inflammatory pain, and regulating the activity of leukocytes and platelets [79]
Resolvin D1 (RvD1)	2 ug/kg	RvD1 reduces macrophages by >50% in adipose tissue (F4/80^+^CD11c^+^ in a male leptin receptor-deficient (db/db) mice) thought ALX/FPR2 lipoxin receptor [80].
Resolvin D3 (RvD3)	10 ng per mouse	Enhances human macrophage efferocytosis and reduces human platelet-PMN aggregation in mice’s model of E. coli peritonitis [81]
Resolvin D4 (RvD4)	1 to 100 nM	Stimulates whole-blood neutrophil phagocytosis of Escherichia coli. RvD4 increased bone marrow macrophage efferocytosis of neutrophils [82]
Resolvin D5 (RvD5)	0.1–1 μg per mouse	Reduce granulocyte infiltration preventing local and systemic inflammation in an intestinal ischemia/reperfusion model and decrease neutrophil counts in a zymosan peritonitis mice model [83].
Resolvin D6 (RvD6)	----	Promotes corneal wound healing and restores corneal innervation after injury mice model and its present in mouse tears [84].
Protectin D1 (PD1)	10 ng	Reduces of PMN and leukocyte infiltration into the inflammatory exudate, and also limits the expression of cytokines of this type, such as IL-6 in a murine model of peritonitis. In addition, PD1 decreases cell damage and, in turn, promotes tissue recovery [76].
Neuroprotectin D1 (NPD1)	100 ng per mouse	PD1 has been studied in models of stroke-mediated brain damage, which is defined as brain lesions associated with an interruption of blood supply to this organ, and ocular damage where it has a powerful protective action on the retina and the brain, giving its neuro-protective character [79].
Maresin 1 (MaR1)	4 to 25 μ/Kg (mouse model) 1 and 10 nM (human in vitro) 1 μg per mouse	In the in vivo and in vitro administration, a very low concentration of MaR1 was able to decrease PMN infiltration and increase phagocytosis of apoptotic PMNs (efferocytosis), which relate to a shortening of the resolution phase of inflammation and restoration of homeostasis, that also protects the remaining cells, that are less exposed to oxidative stress and, therefore, are able to maintain homeostasis [87]. Also, MaR1 increased the proportion of Tr1 cells (CD3^+^, CD4^+^, CD49b^+^, which are crucial in maintaining tolerance to self-antigens) and increased in ~ 50% the percentage of this lymphocyte subset producing IL-10 in a mice model of experimental autoimmune encephalomyelitis (EAE) [89].
**2. Role in macrophage polarization**		
**SPM**	**Doses**	**Effects**
SPMs		Historically, SPMs have been characterized as molecules with anti-inflammatory and resolving characteristics through mechanisms such as reduced inflammatory infiltration, decreased trans-epithelial migration of PMNs, decreased expression of proinflammatory cytokines and increased efferocytosis. This has been related to an improvement in the resolution phase of inflammatory processes and restoration of general homeostasis; thus, SPMs likely act on the inflammatory environment by promoting macrophage polarization towards the anti-inflammatory M2 phenotype [87,89].
RvE1–RvD1	0.1, 1, 10, and 100 nM	In studies of lung inflammation induced by nanomaterials in a murine model, there is a temporal correlation between the endogenous RvD1 and RvE1 peaks and the polarization towards an M2 anti-inflammatory macrophage phenotype [99]. RvD1 administration increased phagocytic activity, attenuating ROS production, a process typically associated with the M1 pro-inflammatory phenotype [100].
MaR1	10 nM 100 ng	MaRs not only is produced by human macrophages, but it also shifted macrophage phenotype from a CD54^+^CD80^+^ M1 to a CD163^+^CD206^+^ M2 phenotype [101]. M1 macrophage differentiation while promoting M2 macrophage differentiation in LPS-stimulated mice, inhibiting an CD38+CD80+CD86+iNOS+ M1 macrophage differentiation promoting a CD36+CD163+CD206+Arg-1+ M2 phenotype [103].
PD1	100 nM	In the context of PD1, it was not possible to find assays related to macrophage polarization activity, although Ma et al. demonstrated that PD1 inhibits inflammatory cell death or piroptosis in macrophages, improving survival in LPS-induced sepsis inflammatory model [104].

CD, cluster of differentiation; IFN-γ, interferon-γ; IL-1β, interleukin-1β; LPS, lipopolyssacharide; PMN, polymorphonuclear cells; TNF-α, tumor necrosis factor-α; ----, no doses informed.

Regarding the participation of SPMs in macrophage polarization and their role in chronic liver disease, Kang et al., in a murine model of hepatic inflammation induced by ischemia-reperfusion, where an M1 polarization of the KCs is favored, demonstrated the pro-resolving and anti-inflammatory effects of RvD1 through the activation of ALX/FPR2 pathways [106]. In particular, RvD1 (15 μg/Kg) promotes both M2 polarization (Arg1, Cd206, and Mst1r) and spherocytosis performed by KCs at 24 h of reperfusion. In addition, the administration of RvD1 modified the inflammatory infiltrate, decreasing the population of neutrophils present in the area after 24 h of reperfusion [107]. In the same line, RvD1 and RvD2 has in vitro positive effects on hepatocellular carcinoma cells and/or tumor associated macrophages, connected with M2d polarization [108,109]. Taken together, these results suggest that RvD1 stimulates the pro-resolving functions of the M2 phenotype and inhibits the proinflammatory character of the macrophage M1 phenotype (Table 2), in models of ischemia-reperfusion and hepatocarcinoma, respectively. Based on the aforementioned studies, it is also possible to highlight the relevance of KCs in the development and resolution of liver diseases, since these are the main agents of PMN phagocytosis and their depletion suppresses the beneficial effects generated by RvD1 [106]. In the case of RvE1, at liver level, data reported are not enough conclusive and there is no information in respect to the role of this Rv over macrophage polarization. For example, Pohl et al. did not find any positive results when RvE1 was administrated in a Non-Alcoholic Steatohepatitis (NASH) rodent model [109]. The hepatic expression of the macrophage marker F4/80 and the inflammatory mediators TNF-α and CCL2 were not altered at doses of 1.2 ng/g body weight [109]. But according to Kuang et al., 10 μg/kg of RvE1 prevent concanavalin A-induced liver injury and the progression of hepatitis to liver cancer in mice through inhibition of inflammatory cytokine secretion and NF-κB/AP-1 activity [110]. Contrarily, in the study of Rodriguez et al., the administration of 100 ng/kg of RvE1 to rats was not enough to induce NF-κB nuclear activity in a fibrosis liver model (diethylnitrosamine induction) [111]. Also, Qiu et al. found that RvE1 at 100 ng/animal decreased the levels of TNF-α in a Schistosoma japonicum-liver fibrosis induction [112]. Although most of the studies inform that RvE1 could have hepatoprotective activities (depending on the doses), there is not enough information to indicate that these benefits are due to macrophage polarization.

From most of all SPMs, MaR1 appears to have the most potent activity against liver injury even at very small doses. It has been shown that the administration of MaR1 can induce cell division, promoting cell cycle and proliferation, undertaking the characteristics of macrophage M2 [96]. For example, in an ischemia-reperfusion liver injury model in rats, the administration of MaR1 at 4 ng/g body weight demonstrated positive effects on the modulation of primary M1 cytokines (such as TNF-α and IL-6), which are generated by KCs during reperfusion, stimulating liver tissue regeneration [95] In another acute liver injury model, Zhang et al. (2020) showed that MaR1 at 50 or 100 ng/kg reverts inflammatory signs of liver noxa induced by concavalin-A (ConA). The authors revealed that MaR1 reduced mortality caused by ConA due to a reduction in ROS levels and NF-κB activity in liver macrophages, which would indicate an inhibition of the M1 polarization pathways [113]. The authors also re-assayed in macrophage RAW264.7 cells, where MaR1 promoted its apoptosis and the M2 phenotype, with the limitation that the cell line used would only reflect the behavior of the macrophages and not necessarily emulated the Kupffer cell activity over CLD.

As mentioned above, the effects of MaR1 in liver tissue would occur through the stimulation of RORα, a ligand-dependent transcription factor that regulates lipid metabolism and inflammation [95]. Its activation, induced by MaR1, leads to the polarization towards M2 in liver macrophages and improved NASH symptoms, also generating positive feedback on MaR1 biosynthesis by inducing 12-LOX [95]. In a study by Han et al., MaR1 induced M2 switch in liver macrophages, by increasing the CD206^+^/CD80^+^ ratio and enhancing the expression of Klf4, Arg1 and Cd163 as M2 markers, all of that under time and dose dependency [102]. Reinforcing the above findings, Yang et al. demonstrated that MaR1 does not promote macrophage polarization in a galactosamine model of acute-liver injury; however, it can suppress the activation of NF-κB and the signaling of the inflammasome NLR family pirin domain containing 3 (NLRP3), which is the signaling via of piroptosis [114]. Moreover, the M2 protective effects could be related to nuclear factor erythroid 2-related factor 2 (Nrf2) upregulation [114]. The fact that the resolutive and hepatoprotective effects of MaR1 are determined from two important factors—(i) the M1 to M2 polarization related to a decrease in NF-κB activation in liver macrophages, but also in the parenchyma; and (ii) the antioxidant response activated via Nrf2 [73,103,114]—conclusively supports the contention that MaR1 is a potent regulator of the molecular machinery in KCs. It is still necessary to understand the role of this SPM in the hepatocytes and how MaR1 connects the machinery related to the crosstalk among the most important type of cells in the liver (hepatocytes, KCs, HSCs, and others), not only to put an end to the damage in chronic liver diseases, but also to use it as a potential regenerative molecule. At present time, however, (i) the complex etiological factors for CLD and the large heterogeneity of KCs challenge the transfer of the current knowledge into the development of macrophage-targeted therapies for CLD; (ii) most of the previous studies on the roles and mechanisms of hepatic macrophages in CLD were carried out in rodent models, thus requiring future research into the applicability of these findings to humans; and (iii) considering that the precise functions and control mechanisms of liver macrophage subclasses in humans are limited, it is a major concern to just focus on the study of pathogenic phenotypes rather than the physiological and resolutive macrophages [115,116,117,118].

## 9. Conclusions

In conclusion, this comprehensive text discusses the intricate relationship between chronic liver diseases, macrophage polarization, and the role of omega-3 fatty acids and their specialized pro-resolving mediators (SPMs). CLDs such as cirrhosis and non-alcoholic fatty liver disease (NAFLD), are characterized by inflammation and fibrosis, where macrophages play a crucial role. The M1 and M2 macrophage phenotypes represent pro-inflammatory and anti-inflammatory states, respectively, and their balance is crucial in the progression and resolution of liver diseases. The shift towards the anti-inflammatory M2 phenotype is associated with improved resolution of inflammation, tissue repair, and reduction in oxidative stress, which are all critical factors in chronic liver diseases. SPMs, such as resolvins (Rvs), protectins (PDs), and especially maresins (MaRs), play a pivotal role in regulating macrophage polarization and inflammation resolution. RvR1, RvR3, PD1, and MaR1 promote the M2 phenotype while inhibiting the M1 phenotype, contributing to tissue repair and resolution of inflammation in various inflammatory diseases, including CLDs. The evidence suggests that they could be promising as therapeutic interventions to modulate macrophage polarization and resolve inflammation, but further studies are needed to fully elucidate the molecular mechanisms underlying the interactions between omega-3 fatty acids, SPMs, and macrophage polarization, and to explore their clinical potential in managing CLDs.

## Figures and Tables

**Figure 1 ijms-24-15528-f001:**
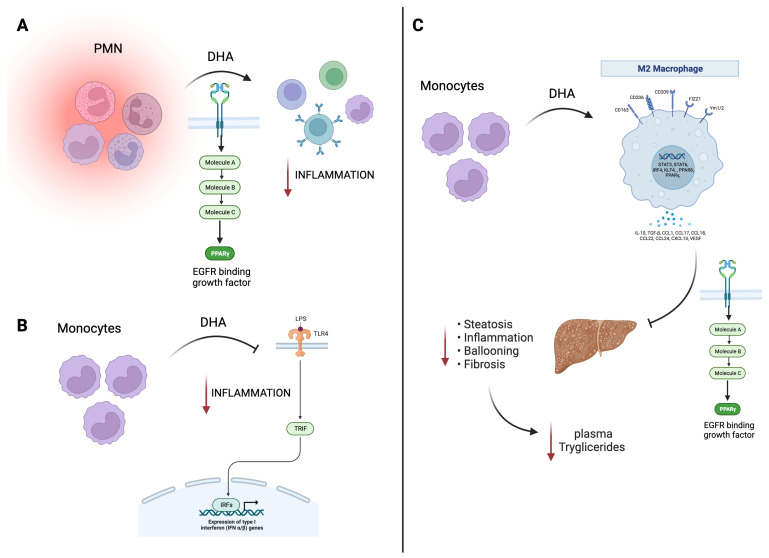
Beneficial effects of EPA/DHA on immune and cardiac models. (**A**) DHA treatment in PMN cells promotes PPAR-γ activation with a decrease in the inflammatory response. (**B**) In murine monocytes, DHA inhibits TLR4 signaling, also translating this to a decrease in inflammation. (**C**) In macrophages, DHA, through PPAR-γ, promotes an hepato-protecting effect decreasing inflammation, steatosis, and fibrosis, related to a decrease in plasma triglycerides. Black arrow informs a activation or inhibition of a pathway; Red arrow is related to negative effects. EGFR, endothelial growth factor receptor; EPA, eicosapentaenoic acid; DHA, docosahexaenoic acid; LPS, lipopolyssacharide; PMN, polymorphonuclear leukocytes; PPAR, peroxisome proliferator-activated receptors; TLR4, toll-like receptor-4; TRIF, TIR-domain-containing adapter-inducing interferon-β; IFN, interferon; STAT, signal transducer and activator of transcription; TGF-β, transforming growth factor beta; VEGF, vascular endothelial growth factor.

**Figure 2 ijms-24-15528-f002:**
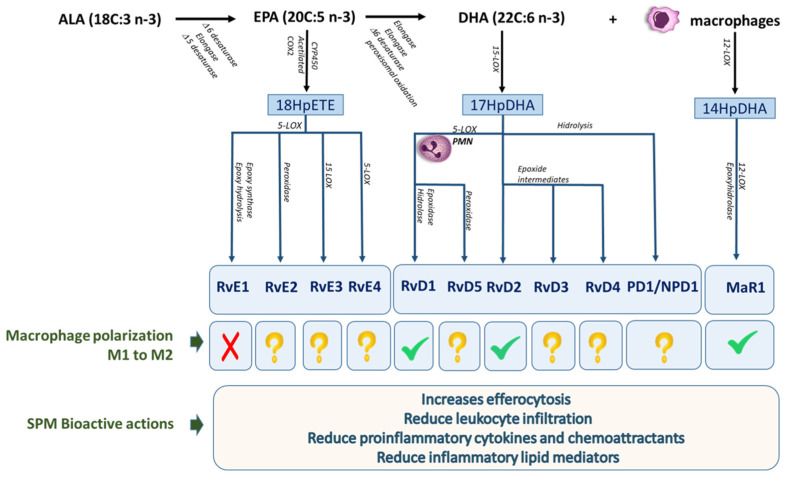
Specialized pro-resolving mediators (SPM): biosynthetic pathways and macrophage polarization activities. E-series resolvins, D-series resolvins, protectins, and maresins are illustrated. ALA, alfa-linoleic acid; EPA, eicosapentaenoic acid; DHA, docosahexaenoic acid; LOX, lipoxygenase; COX, cyclooxygenase; PMN, polymorphonuclear cells.

**Table 1 ijms-24-15528-t001:** Protective functions of omega-3 related to inflammatory and macrophage modulation.

Omega-3	Mechanism	Effect	Model
**Clinical Trials Analysis**			
A	Decrease in T helper 2/T helper 1 chemokines	Lower macrophage-derived chemokine/interferon-inducible protein	Cord plasma from newborns from pregnancy-related depressive mothers (prenatal supplementation [55])
A	Decrease levels of sCD63 *	Decrease in triglycerides, waist to height ratio and waist circumference	Type 2 diabetic patients [56]
A	Modulation of Wnt/ beta-catenine pathways **	White adipose tissue downregulation of inflammatory pathways with less macrophage infiltration	Obese subjects [57]
A	Enhance of CD54 macrophages ***	Less T CD8^+^/T CD4^+^ after immune challenger and greater production of IL-10 (an anti-inflammatory cytokine)	Children (ages 5–7 years), who had low intakes of DHA [58]
A	Less macrophage in atherosclerotic plaques	Improvement of atherosclerotic plaque morphology (tin fibrous cap)	Patients awaiting carotid endarterectomy [59]
**Animals model**			
A	Attenuated the development and destabilization of atherosclerotic plaques and reduction in TLR4	Suppressed atherogenesis	Apolipoprotein E-deficient (ApoE^-/-^) mice [60]
A	Decreased of TNF-α, MCP-1, TGF β, and arginase 2; this last one is a marker of pro-inflammatory macrophages	Reduction in aneurism formation and macrophage infiltration	Abdominal aortic aneurysm (AAA) animal model [61]
C	increased levels of resolvin D5, protectin DX, and maresin 2 in the mouse skin	Decrease proinflammatory cytokines altered psoriasis macrophage phenotypes and lipid oxidation, modulating psoriasis skin inflammation	K14-Rac1V12 mouse model [62]
A	Induction of Nrf2 signaling	Decrease proinflammatory cytokines, iNOS and COX-2	Nrf2 knockout (^-/-^; KO) mice [63]
B	Down-regulation of NF-κB activation and regulated genes	Inhibited tubule-interstitial injury and the infiltration of macrophages into tubule-interstitial lesions	Thy-1 nephritis model [64]
A	reduce TNFα and caspase-3, and could increase splenic GSH Bcl-2. Restoring macrophages, and B- and T- lymphocytes.	Decrease the Methotrexate-induced histopathological injury	Methotrexato-induced splenic suppression on Sprague-Dawley rats [65]
A	Reduced pro-inflammatory macrophages	Promoted wound closure by accelerating the resolution of inflammation	Wound healing in *db*/*db* mice [66]
A	Reduction in hepatic SREBP-1c and enhancement of PPARγ nuclear receptor	Lower concentration on plasma lipids, triglycerides, and liver lipid content. Enhance of endothelial function	HFD in hamsters [67]

A: EPA + DAH; B: EPA; C: EPA or DHA. * Macrophages activation marker; ** Pathway related to genes involved in adipogenesis; *** Co-stimulatory molecule on antigen-presenting cells that facilitates MHC-restricted immune response. EPA, eicosapentaenoic acid; DHA, docosahexaenoic acid; TLR4, toll-like receptor-4; TNF-α, tumor necrosis factor alpha; MCP-1, monocyte chemoattractant protein-1; TGF-β, transforming growth factor beta; Nrf2, nuclear factor erythroid 2-related factor 2; NF-κB, nuclear factor-kappa B; GSH, glutathione; SREBP, sterol regulatory element binding proteins; PPAR, peroxisome proliferator-activated receptors; iNOS, inducible nitric oxide synthase; COX-2, cyclooxygenase-2; HFD, high fat diet.

## Data Availability

Not applicable.

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
