# Peer review of "Omega-3 Lipid Mediators: Modulation of the M1/M2 Macrophage Phenotype and Its Protective Role in Chronic Liver Diseases"

_ijms, 2023, doi:10.3390/ijms242115528_

Round 1

Reviewer 1 Report (Previous Reviewer 2)

The Authors have addressed all my concerns and I have no further comments. 

As far as I am concerned, the manuscript is now acceptable to be published.

Author Response

Thanks

Reviewer 2 Report (Previous Reviewer 1)

In this review, the authors review the existing literature on how omega-3 lipids modulate the phenotypes of M1/M2 macrophages and play a protective role in chronic liver diseases.

The article could be accepted in this form by correcting the bibliographic citations that are in different formats. (line 247). In table 1 the meaning of A, B, and C should be in the table footer. Sometimes reading this manuscript tends to be cumbersome and difficult to follow. Authors are recommended to use tables or more images that try to explain the narrated text.

Author Response

We have prepared a revision R2 of our review article entitled “Omega-3 Lipid Mediators: Modulation of the M1/M2 Macrophage Phenotype and its Protective Role in Chronic Liver Diseases”:

1.- correcting the bibliographic citations that are in different formats.

Answer: The 119 references of the work were revised, and 16 of them were put in the correct format (numbers 6, 16, 30, 46, 48-52, 55, 57, 59, 88, 116-118). Refs 83 and 88 were modified for more relevant and updated information.

2.- In table 1 the meaning of A, B, and C should be in the table footer.

Answer: The meaning of A, B and C, was now located in the footer of Table 1, in page 9.

3.- Sometimes reading this manuscript tends to be cumbersome and difficult to follow. Authors are recommended to use tables or more images that try to explain the narrated text.

Answer:  A relevant part of what is contained in the manuscript related to SPM was improved, generating Table 2, which allows a more fluid reading of the writing. 

Round 2

Reviewer 2 Report (Previous Reviewer 1)

The authors have responded to all questions and the manuscript has improved substantially.

This manuscript is a resubmission of an earlier submission. The following is a list of the peer review reports and author responses from that submission.

Round 1

Reviewer 1 Report

The authors carry out a review of the main omega-3 lipid mediators and their role in the macrophage phenotype in chronic liver disease. The review carried out requires a critical and in-depth reading to specify, focus and direct the reader to the objective of the work. Reading at some points could be simplified by tables that summarize the content.

Some issues should be addressed:

 1. The objective should be clarified in both the abstract and the introduction.

2. The authors should update the global health estimate indicated in the introduction.

3. KC, should be indicated the first time it appears in the text.

4. Section 2. hepatitis B or C infection…..virus infection?

5. Section 3. Reduce and focus.

6. Throughout the review, it must be indicated whether it is murine or human.

7. Section 4. Under milieu stimuli….indicate references; Due to the M1/M2 equilibrium…. Indicate bibliography.

8. The revision really begins in section 5 of this manuscript, so it should be reduced and everything above so as not to lose the focus of the work.

6. PUFAs?

7. Table 1 should indicate more sociodemographic parameters (sex, age, sample size, ethnicity...) of the populations studied, this table is a key point of the review. The first column of the table is similar in all rows, perhaps the authors should unify. All abbreviations included in the table should be in the table footer.

8. The references in table 1 should be integrated into the list of references at the end of the manuscript.

9. Figure 1. Abbreviations contained in the figure?

10. Section 6, it should be focused

11. Section 7. Research in recent years…. Insert bibliography.

12. Titos et al. year??

13. Dalli et al. year?

14. Figure 2. Give the figure a title. It should be self explanatory.

15. All abbreviations should be cited the first time in the text.

16. The conclusion of the work should be more specific to the types of mediators reviewed.

Reviewer 2 Report

In this review paper Videla et al. discussed the role of omega-3 fatty acids in the modulation of M1/M2 macrophage phenotype.

There are several limitations of this study which should be mentioned. 

1. In the Introduction section Authors should introduce the aim of the review.

2. The manuscript often lacks clarity. Therefore, paragraphs should be reorganized in a logical order: chronic liver diseases - inflammation - macrophage polarization - balance of M1 and M2 macrophage phenotypes - omega 3 role on the shift towards the anti-inflammatory M2 phenotype - conclusion derived from the studies described in this review.

3. When Authors describe the role of EPA and DHA on M1/M2 macrophage polarization (pag. 4), they should report the % of the fatty acids and discuss more details about the treatment.

4. Results reported in table 1 should be described in the text in a critical manner.

5. It is strange that in the same article there are “table reference” and references.

6. A critical comparison between clinical studies and experiments carried out on animal models should be introduced in the text.

7. The focus of the article is the role of M1/M2 macrophage phenotype on liver diseases. Therefore, a figure describing the effects of EPA/DHA on immune and cardiac model is confounding.

8. Often pieces of text are written in italics.